# Epigenetic Regulation of Fungal Secondary Metabolism

**DOI:** 10.3390/jof10090648

**Published:** 2024-09-13

**Authors:** Yufei Zhang, Wenbin Yu, Yi Lu, Yichuan Wu, Zhiwei Ouyang, Yayi Tu, Bin He

**Affiliations:** Jiangxi Key Laboratory of Natural Microbial Medicine Research, College of Life Sciences, Jiangxi Science & Technology Normal University, Nanchang 330013, China; zyf40488126zyf@163.com (Y.Z.); y15779358832@163.com (W.Y.); m13616844168@163.com (Y.L.); wuyc991025@163.com (Y.W.); echo_oyzw@163.com (Z.O.)

**Keywords:** fungi, secondary metabolism, epigenetic, cross-regulation

## Abstract

Secondary metabolism is one of the important mechanisms by which fungi adapt to their living environment and promote survival and reproduction. Recent studies have shown that epigenetic regulation, such as DNA methylation, histone modifications, and non-coding RNAs, plays key roles in fungal secondary metabolism and affect fungal growth, survival, and pathogenicity. This review describes recent advances in the study of epigenetic regulation of fungal secondary metabolism. We discuss the way in which epigenetic markers respond to environmental changes and stimulate the production of biologically active compounds by fungi, and the feasibility of these new findings applied to develop new antifungal strategies and optimize secondary metabolism. In addition, we have deliberated on possible future directions of research in this field. A deeper understanding of epigenetic regulatory networks is a key focus for future research.

## 1. Introduction

Fungi, as microorganisms, are distributed all over the world, and its presence is detectable in any environment [1]. It has great potential for synthetic biochemistry applications, especially in the generation of secondary metabolites. Fungi are one of the most important producers of secondary metabolites in nature. They are involved in a wide range of ecological functions and are capable of interacting with each other. In general, secondary metabolites are controlled by clusters of biosynthetic genes, and differences in the degree of gene expression can lead to differences in the amount or type of secondary metabolite produced [2,3,4]. Secondary metabolites, as a complex biosynthetic process, are regulated by different aspects, including the co-regulation of a wide variety of enzymes and transcription factors [5]. Secondary metabolites are synthesized and have a wide variety of biological activities, such as antioxidants, cytotoxicity, anti-microbial agents, etc., which play an important role in medicine, foods, and agriculture [6,7,8]. Marine fungi, as a large group of fungal species, have abundant natural products that can be produced and identified, which has been widely noted by researchers. Of note, marine fungi can produce various types of compounds, such as terpenoids, peptides, polyketides, and alkaloids [9]. Soil fungi, as a complex fungal system, can regulate the health of ecosystems and plants. Its applications in secondary metabolite production and industrial production are also very common, and not only that, but they also play an important role in ecosystems by combatting root pathogens and the onset of early droughts [10]. Filamentous fungi are an important group of fungi that produce pigments, which can be used in a wide variety of industrial applications, such as textiles, cosmetics, and food additives. The pigments produced by filamentous fungi can be used in industrial applications, which are more environmentally friendly and in line with the concept of green production [11]. The presence of toxins in molds may have a direct impact on human health and, in severe cases, may lead to the development of cancer or leukemia by inducing specific biological pathways [1]. There is also the currently studied genus *Trichoderma* spp., which has been recognized as the most promising genus in biological control as it can be used as an alternative to traditional chemically synthesized fungicides. In addition, it’s high reproduction rate, short reproduction time, and specific targeting sites make it suitable for synthesizing biocontrol agents against plant pathogen forms. It has been studied that as many as 25 biocontrol agents have been formed from genus *Trichoderma* spp. and have been used to control many plant fungal diseases. Biocontrol agents for plant diseases are gaining ground as viable alternatives to synthetic pesticides, and these biocontrol agents are considered to have a higher level of safety and minimal negative environmental impacts [12]. For example, some antibiotic secondary metabolites produced by *Trichoderma* spp. fungi, such as aromatic compounds, polyketides, and volatile terpenoids, can be effective as biocontrol agents [13]. Biological control strategies for plant-parasitic nematodes can be an effective alternative to toxic chemical insecticides. Endophytic fungi reduce plant-parasitic nematode attack through parasitism, nematode paralysis, antibiotic production, and spatial competition [14]. *Beauveria* spp. and *Metarhizium* spp. i.e., they can act as endophytes, capable of promoting plant growth and resistance, and as entomopathogens to control pest populations. This dual function highlights the potential of entomopathogenic fungi as innovative and environmentally friendly alternatives to chemical pesticides in pest management [15]. The co-culture of fungi with fungi induces the biosynthesis of secondary metabolites to defend against injury due to competition for environmental resources, thus inducing the synthesis of a few novel natural products. Therefore, in addition to secondary metabolite synthesis performed on individual strains, there are synthetic strategies for co-culturing two fungi to synthesize secondary metabolites. Co-culture of two fungi of marine origin, *Aspergillus sclerotium* and *Penicillium citrinum*, can produce two new toxin compounds, aluminum neo-hydroxyaspergillotoxin and ferrous cyano hydroxyaspergillotoxin. The former showed significant selective cytotoxicity against the human lymphoma U937 cell line and was able to reduce the growth of *Staphylococcus aureus*. This is also a novel strategy to synthesize more natural products [16]. In addition to this, fungal-bacterial co-cultures are potential techniques for the production of active secondary metabolites. The total ethyl acetate extract of the co-culture was able to inhibit Klebsiella pneumoniae. Isolated and purified 11-octadecenoic acid, 2,4-di-tert-butylphenol, 2,3-butanediol, and 9-octadecenamide were found to completely inhibit the bacteria after antibiotic susceptibility testing. The bacteria showed different responses under chemical stress and were able to induce fungal secondary metabolites with antimicrobial activity [17].

Microbial secondary metabolites can play a wide range of important roles, but there are still many novel microbial secondary metabolites that have not yet been discovered, and at the same time, the production of natural microbial secondary metabolites is very low, so there is now an urgent need for us to find a strategy to solve the two problems mentioned above. Currently, epigenetics is one of the most striking fields in biology, offering new research strategies for solving current intractable biological problems [18]. Epigenetic inheritance does not usually alter the underlying DNA sequence but is able to influence the biological characteristics of organisms through gene expression and protein synthesis [19]. In fungi, epigenetic regulators have become major players in the regulation of fungal secondary metabolism by enabling the expression of non-expressed or low-expressed genes, activating clusters of biosynthetic genes, and thereby generating novel natural products that have not been seen before [20]. Some endophytic fungi are producers of biologically active metabolites, but under-expression of biosynthetic pathways occurs under conventional culture conditions. However, since the entire biosynthetic pathway is regulated by natural epigenetic regulators, the production of secondary metabolites can be greatly increased, and their functional properties, such as antimicrobial and antioxidant properties, can be improved [21]. The use of small-molecule epigenetic modifiers enables the development of overexpressed polyketide synthase (PKS) gene strains that produce genetically stable mutant strains and synthesize new secondary metabolites. These epigenetic variants, in turn, differ significantly from wild-type strains in terms of phenotype and secondary metabolite synthesis [22]. The combination of a histone deacetylase inhibitor and a DNA methyltransferase inhibitor enabled the isolation of one new compound and seven known compounds from cultures of the marine fungus *Aspergillus* sp. SCSIOW3, which contributed significantly to the enrichment of secondary metabolite species [23,24]. Overexpression of histone acetyltransferase and spectroscopic experiments on the plant endophyte fungus *Monosporascus eutypoides* revealed the production of two newly discovered secondary metabolites and two known secondary metabolites, increasing the product diversity of the fungal genome [25]. The addition of the histone deacetylase inhibitor-substituted hydroxamic acid (SBHA) to *Arthrobotrys foliicola* medium enabled the isolation of coumarin metabolites, which are derived from arthropods and are capable of antimicrobial, anti-inflammatory, and insecticidal effects [26]. Epigenetic modifications are also capable of mediating the production of potential chemical effectors as defense mechanisms against the external environment, and the synthesis of such metabolites could likewise provide potential therapeutic strategies in cancer treatment [27].

In fact, epigenetics can effectively regulate the amount of fungal secondary metabolite synthesis as well as increase the variety of metabolites. Thus, the scaled-up application of epigenetic means in biosynthesis will contribute to new and vibrant metabolomics in the future.

## 2. Secondary Metabolism in Fungi

Secondary metabolites (SMs) are a vast array of small molecules produced by microbes and plants. These molecules are not essential for the normal growth of an organism but play important roles [28]. Generally, secondary metabolites can be produced by many different metabolic pathways and include antibiotics, cytotoxic and cyto-stimulatory compounds [29]. Unlike primary metabolism, secondary metabolism is usually controlled by factors such as nutrients, growth rate, feedback control, enzyme inactivation, and induction [6,30,31,32]. Marine fungi have been shown to produce a variety of secondary metabolites with various structures and bioactivities, including antibacterial, antiviral, anticancer, and anti-inflammatory characteristics, and have already provided a number of promising leads against MRSA [33]. Deep-sea hydrothermal vents are characterized by high temperature and pressure, lack of sunlight, and low oxygen. Nearby marine microbes have attracted the attention of researchers, who have reported some anti-inflammatory phenazine alkaloids extracted from the yeast-like fungus *Cysticercus laryngalis*, which have anti-microbial and cytotoxic activities [34]. *Aspergillus sydowii*, the causative agent of marine fan coral aspergillosis, is capable of producing structurally diverse secondary metabolites such as alkaloids, polyketones, and sesquiterpenes [35]. In the natural environment, fungi are bound to compete with other organisms for resources. This leads to the biosynthesis of secondary metabolites in fungi for defense purposes. Fungi of the genus *Cytospora* are prolific sources of structurally diverse and biologically significant metabolites, especially capable of producing potent antimicrobial compounds [36]. According to literature reports, endophytic fungi secondary metabolites not only have antibacterial effects but also have herbicidal activity, which such function has been reported in relatively few studies [37]. Root rot of conifers causes significant economic losses and is a major problem in forestry. Compounds isolated from the biocontrol fungus *Macrobacterium* were studied, and from the three compounds isolated, the natural secondary metabolite o-p-acetaldehyde was found to be antifungally active [38]. Secondary metabolites of fungi are not only used in medicine and agriculture but also in cosmetics, food, chemical synthesis, and other fields. For example, kojic acid, a secondary metabolite produced by *Aspergillus flavus* or *Aspergillus oryzae*, is a tyrosinase inhibitor and can be used as a skin-whitening agent [8]. In the past, insect pathogens could be studied as biocontrol agents. Secondary metabolites derived from Hypocrealean entomopathogenic fungi (HEF) can regulate both intra-species and inter-species communication. Similarly, some of these secondary metabolites can be used in medical treatment, and they can be made into health products or drugs through modern pharmaceutical processes [39]. Of note, entomopathogenic fungi synthesize non-volatile metabolites that can act as insecticides by counteracting the immune system of insects. Some fungi of the ergot family can generate a wide range of nutraceuticals as secondary metabolites that can act as anticancer agents, antimicrobials, and modulators of the immune system and the nervous system [39,40,41]. Simultaneous inoculation of arbuscular mycorrhizal fungi and *Methylobacterium oryzae* consistently improves the absorption of macronutrients and micronutrients in red peppers, which relies on the perfect reciprocity of these two beneficial and interacting species. The interaction between the two beneficial species can serve as an alternative to chemical fertilizers and contribute to the use of microbial products in biotechnology [42]. The synthesis of microbial secondary metabolism is generally regulated by the regulatory mechanisms of environmental signaling, transcriptional regulation, and epigenetic regulation [5]. Environmental signaling regulation includes various aspects such as nitrogen regulation, light regulation, and other factors. Fungal cell wall integrity signaling activates the biosynthesis of secondary metabolites. Damage of the cell wall affects fungal melanin production by altering the cell shape and resulting in unbalanced nutrient acquisition [43]. Transcriptional regulation is one of the important modes of regulation in the synthesis of secondary metabolites. The global transcriptional regulator HbxA of *Aspergillus fumigatus* can influence morphogenesis and secondary metabolism. In the absence of HbxA, *Aspergillus fumigatus* conidia are enlarged and show a high germination rate. Therefore, the presence of HbxA reduces conidial production, resulting in fewer *Aspergillus fumigatus* conidia being inhaled during human respiration, thus reducing lung infections [44]. Epigenetics is defined as any heritable change that does not involve a change in the DNA sequence. Modifications are mainly mediated by DNA, RNA, or chromatin [45]. Epigenetic regulation plays an important role in DNA repair, replication, and transcription, mainly including methylation, acetylation, phosphorylation, ubiquitination, and SUMOylation [46]. Epigenetic regulation plays a key role in the metabolism of fungal natural products. Natural metabolites derived from microorganisms can inhibit or activate biosynthetic gene clusters under the regulation of epigenetic means to generate novel secondary metabolites [20].

## 3. Epigenetic Regulation

Chromatin structure is the basis for regulating gene expression. Chromatin is composed of proteins and DNA in which the genetic material of eukaryotes is contained. A nucleosome, the basic subunit of chromatin, consists of approximately 165 bp of DNA that wraps four different core histones [47]. The open structure of euchromatin is associated with transcriptional activity. The tight structure of heterochromatin is associated with transcriptional repression [48]. Post-translational modifications in epigenetics occur at specific proteomes. Modifications are made by way of DNA methylation, acetylation, deacetylation, and phosphorylation, which are involved in the control of DNA expression [49]. In the production of secondary metabolites, epigenetics can be used as a key mechanism to change the expression of genes encoding secondary metabolites to improve the production of secondary metabolites [50]. In particular, epigenetic inheritance is increasing in value as a potential application for disease treatment. In humans, microbial structures and microbial secondary metabolites interact with host cells to maintain homeostasis in vivo, and if the levels of metabolites are elevated, epigenetic effects may be induced, leading to disease [51]. Metabolic diseases, such as obesity, dyslipidemia, diabetes mellitus, and fatty liver disease, are becoming increasingly prevalent in real life, and such diseases that carry a high risk of morbidity and mortality, placing a burden on human society. The epigenetic effects of diet-related gut microbes can be involved in the regulation of metabolic disorders and thus in the treatment of metabolic diseases [52]. Antifungal therapies are an important area of research in today’s medicine, where human fungal pathogens develop resistance through multiple mechanisms. Epigenetic regulation, in which histones are dynamically acetylated and methylated, contributes to the control of medical fungal resistance to cell wall- and cell membrane-targeted antifungal drugs [45]. Against *Plasmodium falciparum*, epigenetic process regulation has also been developed as a promising target for multi-stage anti-malarial drugs [53]. In the epigenetic regulation of growth and the pathogenicity of plant fungal pathogens, persistence in different host ecological niches has been shown to play an important role. Epigenetic modifications of different pathogens in different geographical locations provide information on trends in pathogenesis or pathogenicity [54]. Epigenetic inheritance can still play a role as an important research tool in the application of plant senescence and crop improvement. The main cause of cellular senescence has been recognized as DNA damage over the past 20 years. The altered epigenetic information can have an impact on cellular senescence in organisms, which will play a key role in post-transcriptional gene regulation and, in turn, affect the fate of RNA, and ultimately has an impact on plant developmental senescence [55]. Indeed, epigenetic regulation is not only limited to several self-regulatory modalities to regulate. CRISPR/dCas9 can be used as a tool for gene transcription regulation in combination with epigenetic mechanisms in various research directions. dCas9 binds to DNA methyltransferases and histone deacetylases to control intracellular methylation or acetylation levels to repress or activate target gene expression. In addition, it controls intracellular methylation or acetylation levels to repress or activate the expression of target genes [56]. Chemical epigenetic factor regulation is also an effective means to activate fungal secondary metabolism. For example, some methylation transferase or acetylation transferase inhibitors are added to increase the diversity of active metabolites [57]. The application of chemical epigenetic modifiers to the plant endophyte fungus *Aspergillus fumigatus* was able to significantly alter metabolic profiles, generate a variety of new natural products, and isolate immunosuppressive agents [58]. The endophytic ascomycete fungus *Chalara* sp. *6661* can produce the isofusidienol class of antibiotics. When co-operating with the histone deacetylation inhibitor vorinostat, it is able to produce new xanthones [59]. On the basis of the research and technological development of epigenetics, epigenetic inhibitors have been applied in medical treatment, and some remarkable results have been achieved, particularly with a wide application in cancer treatment [60]. At present, the FDA has approved seven drugs for the treatment of cancer. They are two DNA methyltransferase inhibitors (Azacitidine and Decitabine) and five other histone deacetylase inhibitors (Vorinostat, Romidepsin, Belinostat, Panobinostat, and Chidamide) [61].

Three prevalent epigenetic regulatory mechanisms in fungi are DNA methylation, histone modification, and RNA-silencing systems [54]. The most widely used tool in epigenetics is histone modification. It is the direct modification of N-terminal histone tails. Among others, they include methylation, acetylation, ubiquitination, phosphorylation, SUMOylation, proline isomerization, and ADP glycosylation [62,63]. Epigenetic modifications can occur at many sites in histones (Figure 1) [49]. *Fusarium verticillioides*, *Colletotrichum higginsianum*, and *Aspergillus fumigatus* strains are capable of histone methylation modification and synthetic regulation of the secondary metabolites they synthesize. *Aspergillus terreus*, *Aspergillus niger*, and other strains can be modified by histone acetylation or deacetylation. Histone phosphorylation modifications and ubiquitination modifications are similarly widespread in fungi and play a key role in fungal growth morphology and cytotoxicity (Table 1). The following is a specific description of epigenetic modifications in different species.

### 3.1. DNA Methylation

DNA methylation is a form of chemical modification of DNA, which commonly occurs in bacteria, archaea, and eukaryotes, that could change genetic performance without changing the DNA sequence. DNA methylation is an important epigenetic mechanism involving the silencing of transposable elements (TEs), the stability of the genome, inactivation of the X chromosome, and marking of the genome [80]. DNA methylation plays an important role in eukaryotic gene expression and silencing, cell differentiation, phylogeny, etc. It can cause changes in chromatin structure, DNA stability, and DNA-protein interactions, thus affecting gene expression. Compared with other eukaryotes, the probability of DNA methylation in fungi is relatively low, and there are some differences. DNA methylation of *C. militaris* can induce strain degeneration and have important effects on major metabolic pathways and synthetic pathways [68]. DNA methylation is a stabilizing regulator of gene silencing, and its main regulatory mechanism is to regulate the expression of genetic information by altering the molecular conformation of DNA and the structural properties of chromosomes. The main site of this mechanism is in the CpG island of the regulatory transcriptional region, which in turn affects histone modification and influences the transcription and expression of related genes [81,82,83] (Figure 2B). That is, under the action of DNA methyltransferase, the fifth carbon atom on cytosine in the dinucleotide of genomic CpG island is methylated [84] (Figure 2A). The distribution of DNA methylation is mainly in gene promoter regions, transcription element regions, and repeat sequence and gene transcription regions [85], which play a crucial role in many biological processes [86]. DNA methylation is relatively stable and heritable, playing a central role in the development of mammals, plants, and fungi [85]. DNA methylation targets tandem and interspersed repeats, particularly transposable elements (TEs). The process of DNA methylation occurs in three different sequences: the symmetric CG dinucleotide, the symmetric CHG, where the H can be A, T, or C, and the asymmetric CHH. The probability of methylation at CHH is very low, probably 10% or less. In addition, DNA methylation has also been found on CG residues within gene bodies [87]. Methyltransferase 1 (MET1) maintains CG methylation. Chromosomal methylase (CMT3) maintains CHG methylation. DRM1/2 is a CMT2 and DNMT3 homolog, and it can be used to maintain CHH methylation [88]. The reverse reaction of DNA methylation, DNA demethylation, is a complex process in which many proteins and regulatory factors are involved to maintain the balance between DNA methylation and DNA demethylation [89].

According to the results from bisulfite sequencing, DNA methylation in fungi mainly occurs at the CG site, and plays a role in global re-editing during development [90]. Some members of the broadly conserved DNA methyltransferases (DMTs) are both mediators of the repetitively induced point mutation (RIP) genomic defense system and key players in sexual reproduction [90]. Some features of the RIP sequence trigger de novo DNA methylation, which can promote de novo DNA methylation once the recognition of the RIP sequence is easy [91]. The degree of DNA methylation can be affected by various factors. In *Candida albicans*, there is a direct relationship between the time of exposure to ozone and the degree of DNA methylation. The level of total DNA methylation increased significantly with extended exposure time [65]. The DNA methylation of most fungi is lower in mycelium stage than in conidia stage, but it is higher in *Beauveria bassiana* mycelium. Thus, DNA methylation plays an important role in the division and growth of fungal cells [92]. N6-methyladenosine (6mA) is also a DNA methylation modification, and dsDNA 6mA demethylase was found in the dioxygenase CcTet of the fungus *Coprinopsis cinerea*. It is demonstrated that 6mA demethylation is also present in fungi [93]. So far, 5-methylcytosine (5mC) has been studied more than 4-methylcytosine (4mC) and 6-methyladenine (6mA), which may affect the development of fungal species [94]. *Stachybotrys* is a model for studying DNA methylation and developmental symbiosis, in which the roles of 5mC highlights the importance of epigenetic inheritance. 5mC, a DNA modification that is widely present in organisms, is an important epigenetic regulator that modulates transposable element (TE) gene expression or silencing [95]. One of the mechanisms of DNA methylation is to exert its repressive function by stereoscopic interference with transcription factor-DNA interactions or by recruiting DNA methylation-binding repression complexes to chromatin 1,2 [96] (Figure 2C). The biological nature of transcription factors leads to their reactivation and transcription after being blocked by DNA methylation. DNA methylation plays an important role in maintaining the integrity of the genome by silencing transposons and harmful DNA [97]. In addition, another type of methylation, namely RNA-directed DNA methylation, is a biological process of DNA methylation directed by RNA. In particular, there are highly conserved RNAi pathways in fungi [98]. De novo DNA methylation is mainly dependent on the RNA-directed DNA methylation (RdDM) pathway, which is directed by RNAs to specific genomic sequences [99]. In recent years, it has also been found that DNA demethylation and histone methylation can play a mutual regulation and joint role in epigenetic modification, which had made a great contribution in the epigenetic and transcriptional regulation of fungi. Especially, it’s helpful to understand the whole epigenetic system [100]. As of now, the role of DNA methylation in fungi is not widely used. In the future, the exploration of the application of DNA methylation in fungi must be more in-depth.

### 3.2. Histone Methylation

Histone post-translational modification (PTMs) is a key epigenetic regulatory mechanism. Biologically, histone methylation plays an important role as a type of PTM [101]. Histones are components of chromatin, the nucleosome consists of H2A, H2B, H3 and H4, around which the DNA is wrapped, and PTMs are highly modified to regulate the structure and function of chromatin. Histone methylation is mainly performed on lysine or arginine residues by methyltransferase using S-adenosylmethionine as the methyl donor [102]. Histone methylation is regulated by histone methyltransferases (HMTs) and histone demethylases (HDMs) (Figure 3), which act to add and remove methyl groups from lysine and arginine residues, respectively [103].

Histone lysine residues can be modified by one, two, or three methylations in response to lysin-specific methyltransferases (KMTs). To form monomethylated, dimethylated, or trimethylated lysines (Kme1, Kme2, Kme3) [104]. Many protein methyltransferases catalyze the methyl transfer of S-adenosine-L-methionine (SAM) to residues in the lysine side chain [105]. Histone methyltransferases generally have specific domains, and these specific domains can achieve the effect of adding methyl groups at different sites. For example, there are SET enhancer domains in yeast, animal, plant cells, or there are Dot1 domains in yeast and animals [106]. The SET enhancer domain is responsible for most of the known catalytic activities of KMT and is able to catalyze the transfer of methyl groups from S-adenosylmethionine (SAM) to lysine side chains [107]. In recent years, the research on the two related sites of histone H3 lysine 4 and lysine 36 (H3K4, H3K36), as well as histone H3 lysine 9 (H3K9) and histone H3 lysine 27 (H3K27)have made great progress [108]. For histone methylation, lysine has the optimal chain length for histone lysine methyltransferase catalysis [109]. The lysine side chain is located in the narrow polar channel and correctly pointed to the SAM electrophilic methyl group for efficient nucleophilic attack. However, KMTs with the exception of lysine methylation have limited catalytic capacity [110]. H3K4me and H3K36me are associated with transcriptional activation, while H3K9me and H3K27me are associated with heterochromatinization and transcriptional silencing [111]. Several KMTs and can interact with substrates, so it can be suggested that lysine methylation has a potential enzymatic regulatory mechanism. There are specific experiments that show that trimethylation of histone H3 lysine 4 impairs methylation of histone H3 lysine 9 upon lysine methyltransferase interaction with a substrate [107]. According to the experimental results, the interaction between the acidic patch and the SET8 arginine anchor in the histone plays an important role in the activity of H4K20 monomethylation. Binding of the SET8 arginine anchor site to the acidic patch can determine the orientation of the SET domain on the nucleosome for action by the H4 N-terminal tail [112]. SAM has an interdependent relationship with HMTs. In eukaryotic cells, S-adenosylmethionine (SAM) acts as a co-substrate for methyl donors and needs to be relied upon to support HMTs activity. According to the report of biochemical analysis of HMTs in the experiment, it is shown that the enhancement of methylation state leads to a decrease of the catalytic efficiency of the enzyme. According to the kinetic study of the enzyme, the histone H3 lysine-9 methyltransferase maintains the monomethylation level under the condition of low S-adenosylmethionine [113].

Histone arginine methylation is a pattern of histone modification that can easily regulate the structure of arginine residues and binding interactions with other proteins. This approach involves a variety of cellular and subcellular processes, such as pre-mRNA splicing, DNA damage signaling, and cell signaling [114]. It is a ubiquitous post-translational modification (PTM) in eukaryotes. It is catalyzed by the family of protein arginine methyltransferases (PRMTs). Expression can be activated or inhibited depending on the environment [115]. Arginine methylation in the core histone tail plays an important role in regulating chromatin function. The family of histone arginine methyltransferases (PRMTs) consists of nine members (PRMT1-9) [115,116]. There is an arginine methylation fine-tuning mechanism in PRMTs. For example, arginine methylation of PRMT1 decreased with acylation of H4K5; The enhanced methylation of H4K5 could thus enhance the arginine methylation of PRMT3. The arginine methylation of PRMT8 could be decreased acylation. H4K5 acetylation would promote arginine methylation of PRMT5 [117]. Now, it has been pointed out that non-histone arginine methylation can also be carried out by using histone arginine methyltransferase as a mediator [118]. For example, histone H3 arginine methylation regulates the protein arginine methyltransferase 6 to mediate problems associated with cardiac hypertrophy [119]. For protein arginine methyltransferases (PRMTs), arginine methyltransferases inhibitors were randomly selected in related experiments. In this area, the existence of only one inhibitor has been found for the time being. New inhibitors can demethylate cells, which could be used in medicine, especially for hormone-dependent cancers [120].

Chromatin biology is a rapidly developing research direction in modern biological research. Chromatin biology in fungi can have very good applications in medicine or plant pathology [108]. The repressive histone mark H3K27me3 supports highly conserved genes in fungi. It has a wide abundance range and is one of the most studied post-translational marks in fungi. Allow its responsiveness of gene expression regulation, and in support of facultative heterochromatin aspects play an important role [121]. Histone modifications are also associated with DNA methylation. In the filamentous fungus *Neurospora crassa*, the enhanced DNA methylation is suppressed by the expression of histone H1 [122]. Histone post-translational modifications (HPTMs) are involved in the growth and development of *Aspergillus*, virulence, and biosynthesis of secondary metabolites. As a deep fungal infection caused by *Aspergillus* infection, invasive aspergillosis leads to the morbidity and even death of patients with low immunity and has attracted the attention of researchers. Pathogenic Aspergillus species include *Aspergillus fumigatus*, *Aspergillus flavus*, *Aspergillus Niger*, *Aspergillus geophyta* and *Aspergillus terrestris*. Several HPTMs inhibitors are being developed to target drugs for this fungal disease [123,124]. Crosstalk among protein modifications always occur, such as a protein in the filamentous ascomycetes Fng1 involved in histone acetylation, acetylation and methylation of crosstalk [125]. Some demethylases are relatively independent in the secondary metabolism of some fungi. For example, in the plant pathogenic fungus *Fusarium*, H3K4 demethylase activation is independent, while the *Saccharomyces cerevisiae* methyltransferase Set1 has a chromatin regulation function independent of H3K4 methylation [126]. In recent years, there has been increasing evidence that COMPASS complex mediated H3K4 methylation can regulate fungal development, secondary metabolism, stress response and virulence traits. For example, the growth and virulence of *Cryptococcus* can also be crosstalk between H2Bub1 and H3K4 methylation. In *Fusarium graminearum*, FgSet1 controls H3K4me and plays an important role in the response to cell wall damage agents. In the light of *Candida albicans*, H3K4 methylation can change the expression of specific genes, the way of undermining its resistance [127,128,129]. In the fungus *Neurospora*, dimethylation and trimethylation of H3K27me cause selective gene silencing in eukaryotic development. However, telomere repeats at chromosome ends are critical for subtelomeric H3K27 methylation. Chromosome ends also contribute to the deposition of H3K27me on adjacent chromatin [130]. A food-borne pathogenic fungus widely distributed in nature, the common *Aspergillus flavus*, regulates the production of the secondary metabolite aflatoxin B1 via the apparent genetic regulator SET9 [131]. Histone methylation affecting gene expression has been widely used in fungi, and the construction of its regulatory network is a major challenge for future research.

### 3.3. Histone Acetylation

One of the major classes of epigenetic mechanisms involved in the regulation of gene expression is the acetylation and deacetylation of histones. For example, acetylation and deacetylation of lysine residues in histone tails, which subsequently determines the targeted gene expression [132]. In contrast, high expression of deacetylases disrupts the homeostasis of fine regulation of mesoacetylation in histones and non-histone proteins. This leads to alterations in some of the genes that regulate cells during growth [133]. We know that the acetylation level of histone acetylation is regulated by histone acetyltransferases (HATs) and histone deacetyl transferases (HDACs), which are co-regulated by both classes of enzymes, and that they are able to determine whether the chromatin is tightly packed together or becomes loosely packed, thereby controlling the regulation of transcriptional activation of DNA-binding proteins [134] (Figure 4). It also occurs that this may bring genes closer to or away from transcription factors that continue to affect gene expression [135]. That is, transcriptional regulation of gene expression is dependent on the state of chromatin, and chromatin competence itself is dependent on epigenetic regulation [136]. The molecular mechanism of mammalian Syrian hamster histone acetylation is a key step in the application to social stress-related fear memory. This means that more effective methods can be developed to treat adverse behavioral responses to stressful events. Transcription of DNA is necessary for the development and maintenance of long-term memory capacity and can cause a range of behavioral changes. For example, the inhibition of HDACs enhances long-term memory and HAT inhibition impairs memory [137]. The structural dynamics of biomolecular systems were studied by computer simulations. Although H3,H4,H2A, and H2B all showed tail acetylation, computer simulations showed that acetylation of the tail of histone H4 had the most effective effect on chromatin disassembly, which implies that histone H4 tail acetylation has a more positive effect on binding to transcription factors and gene expression [138]. Acetylation of histone lysine reduces the electrostatic interactions between histones and DNA.H3K23ac and H3K14ac are highly correlated in vitro and in vivo. The natural MORF complex is an H3K23-specific acetyltransferase, then for mass spectrometry, biochemical and genomic analyses, H3K14ac promotes acetylation of H3K23 by the MORF complex and transcriptional activation [139].

Histone acetylation is widespread in eukaryotes and plays an important role, for example, in defense processes during plant–pathogen interactions. When a fungus infects a plant body, e.g., maize, the degree of acetylation of the histones H3 and H4 of the plant concerned becomes higher and goes on to affect the promoter transcription of the genes concerned. However, only a few histone deacetylases are capable of regulating plant defense mechanisms [140]. Histone acetylation modifications act similarly on the neural aspects of animals, just as imbalances in H4 histones can cause the onset and subsequent development of a variety of neurological disorders. Similarly, H4 acetylation could also exist as a possible biomarker of tissue remodeling after spinal cord injury in animals [141]. A secondary metabolite of *Aspergillus terreus*, lovastatin, is an excellent cholesterol-lowering drug. The degree of histone acetylation also has a great influence on the regulation of gene expression in the growth and development of fungi. A high degree of H3K56 acetylation promotes histone-associated DNA during the S phase of cell growth, whereas a low degree of acetylation frees histone–DNA complexes during the G2 or M phase of cell growth [76]. Some histone deacetylases play a crucial role in DNA synthesis and repair, mycelial segregation, and asexual development. The deacetylase HDAC can deacetylate the H4K16 site in yeast, and it affects not only one histone but often simultaneously and indirectly affects the acetylation of other histones. HDAC can also acetylate the H4K16 site in yeast and also indirectly controls the H3K56 site in *C. albicans* and modifies it [142]. *Candida albicans* infections are still a major contributor to fungal infections in humans, and the treatment of *Candida albicans* infections continues to be a major challenge for medical and pharmaceutical communities. There is a significant need for targeted drugs to inhibit *Candida albicans*. Interruptions in the regulation of acetylation and deacetylation homeostatic proteins inhibit *Candida albicans* growth, morphology, and virulence, which leads to the treatment of *Candida albicans* infections [143]. SAS3, as a histone acetyltransferase, plays a very important role in global acetylation regulation as well as transcriptional regulation. While SAS3 deletion in yeast does not lead to significant phenotypic changes, deletion of the SAS3 homologue, MoSas3, in *Saccharomyces cerevisiae* appears to have a very far-reaching effect. Deletion of MoSas3 reduces the level of acetylation of H3K14 by nearly half. Compared to the wild-type mycelium, the defective strain had reduced mycelial growth, inactive strain metabolism, extremely reduced conidial formation, and was rarely able to germinate [144]. Currently, a dual-regulated therapeutic approach is being applied to treat leukemia and fungal infections at the same time, opening new possibilities. The co-regulation of histone deacetylase and JAK2 kinase can demonstrate a good anti-tumor effect, on the one hand, and the treatment of drug-resistant *Candida albicans* on the other [145]. The deletion of Sir2 from the maize black grub fungus plays a role in the pathogenesis of fungal plants and is associated with physiological activity. Sir2 represses genes that express biotrophic developmental processes, and it is able to promote the formation of filamentous fibers in plants in the case of knockdown, whereas in the case of overexpression, it can obtain a substantial reduction in the production of tumors in the plant [146]. *Botrytis cinerea* is able to regulate saprophytic to infectious growth, but this mechanism has not been studied in depth, where deletion of the Sas2 gene reduces histone H4 acetylation, which significantly reduces the virulence effect and the oxidative stress effect [147]. Over time, the generation of acetyltransferase complexes has now emerged, and the acetyltransferase complex (SAGA) consists of Spt-Ada-Gcn5. Its HAT module plays a unique role against histone H3K9 acetylation, virulence effects, drug resistance, and oxidative stress. Separate deletions are introduced to different acetyltransferases to observe different biochemical phenomena. Not only were the above mentioned effects observed, but the synthesis of biofilm by the HAT module was also impaired [148]. Rpd3 histone deacetylase deacetylates H3 and H4 histone lysine residues and has been implicated in the growth, virulence, and other effects of the fungal insect pathogen *Beauveria bassiana*, with important insights into the regulation of transcription, translation, and post-translational modifications in filamentous fungi. Rpd3 can lead to either hyperacetylation or hypoacetylation of histone lysine [149]. In the entomopathogenic fungus *Metarhizium robertsii*, deletion of histone acetyltransferases following the deletion of epigenetic regulators led to the activation of silenced factors regulating secondary metabolites, ultimately leading to the generation of 11 new naturally synthesized products. This study gives a new approach to the discovery of new small molecule natural synthesis products [150]. In addition to the combined therapy performed by drugs and epigenetics described above, there is a new strategy for the treatment of *Candida albicans* using a combination of heat shock proteins (Hsp90) and histone deacetylases. Histone deacetylase inhibitors are capable of upregulating the genes of ERG and CDR, mediated by Hsp90, thus enhancing the therapeutic effect of azoles, and they are able to play a much more potent role in the treatment of *Candida albicans* than drugs that are not co-regulated [151]. Catalase protects cells from oxidative damage, and in the filamentous fungus *Neurospora crassa*, catalase serves as a conserved mechanism for the transcriptional regulation of catalase through histone acetyltransferase Gcn5 and a series of stringent regulations [152]. Deletion of the histone deacetylase gene MrHos3 in *Monascus ruber* activated the expression of a number of other related genes in the biosynthetic gene cluster, which was evident from the tendency to increase the level of histone acetylation of H3K9 and H4K12, as well as H3K18. These changes regulate the synthesis of secondary metabolites [153]. The KERS chromatin complex in *Aspergillus flavus* controls its growth, secondary metabolism, and pathogenicity. The KERS complex consists of KdmB-EcoA-RpdA-SntB, and KdmB and RpdA regulate their respective clusters of secondary metabolic biosynthesis genes and in particular play an important role in the regulation of secondary metabolites, i.e., aflatoxin production. RpdA can also act on the level of H3K14 acetylation to co-regulate the production of secondary metabolites [154].

Histone acetylation modifications have been widely explored in fungi. Such modifications play important roles in chromatin structure, biological processes, and disease development. In the future, with further research, we expect to better understand the mechanism of this modification and develop more effective antifungal therapeutics.

### 3.4. Other Epigenetic Regulation

There are other epigenetic modifications besides histone methylation or histone acetylation that we often see, such as ubiquitination, phosphorylation, SUMOylation, proline isomerization and glycosylation [155]. Histone phosphorylation, a more prevalent post-translational modification, plays a role in the coordinated control of gene expression in dynamic epigenomic regulatory forces. Protein kinase MSK1 enhances in vitro regulation of H3S10 phosphorylation and H3S28 phosphorylation and enhances gene expression at nearby promoters [156]. Histone phosphorylation occurs mainly at the N-terminal position of serine (Ser) and threonine (Thr), and is associated with transcriptional activation, DNA recombination repair, and chromosome differentiation. H3 histones are the most frequently modified and are also found in H2A histones. Histone phosphorylation is associated with mitophagy and can control chromosome divisions in plant cell. H3 histone phosphorylation is associated with the activity of the filament, and is capable of controlling chromosome divisions in plant cell [157]. According to the literature, histone phosphorylation has effects on other epigenetic modifications, such as the phosphorylation of histones by certain cleaved kinases, which attenuates arginine methylation, which in turn regulates gene expression. Knockdown of TRPM6 or kinases leads to alterations in histone S/T phosphorylation, and M6CK is directed to specific sites for histone acetylation, which in turn attenuates local arginine methylation to regulate transcription [158]. NatD acts as an epigenetic regulator of lung cancer invasion, and its deletion represses the transcription of the Slug gene. NatD is able to antagonize serine phosphorylation by mediating the acetylation of histone H4 to promote epithelial-to-mesenchymal transition (EMT) in lung cancer. Gene transcription and epigenetic regulation interact to find potential targets for lung cancer therapy [159].

Ubiquitin contains 76 amino acids. Its attachment to the target protein facilitates an enzymatic reaction that produces an isopeptide bond, and this isopeptide bond is able to connect the hydroxyl end of the ubiquitin protein to the amino group at the end of the lysine residue of histone H2A. Deubiquitinating enzymes (DUBs) dynamically regulate the entire ubiquitination process, including the N-terminal and C-terminal sites of histone H2A, and play an important role in coordinating the DNA damage response [160]. H2B monoubiquitination is also capable of crosstalk with other histone modifications that play an important role in regulating adaptation to environmental changes in plant growth and development. These modifications can fine-tune transcription and ensure developmental plasticity [161]. The homologous recombinant knockout of AaBRE1 in the plant leaf spot pathogenic fungus *Alternaria alternata* leads to the deletion of histone H2B monoubiquitination, which in turn is also able to reduce histone H3 lysine 4 trimethylation, resulting in the formation of macromolecular complexes, the process of cellular metabolism as well as the process of mycelial growth and conidial formation and pathogenicity play an important contribution [78]. The ubiquitin ligase E3 has a unique target protein recognition ability, and this function has been linked to plant pathogenicity. The deletion of two genes, VdBre1 and VdHre1, impairs fertility and penetrance. Whereas DVBre1 controls lipid metabolic pathways, its mutants could reduce levels of histone ubiquitination and histone H3K4 trimethylation, which ultimately controls toxic effects [162]. Histone ubiquitination also affects transcription and chromatin structure at sites near broken double strands, thereby affecting double strand breakage and repair. Ubiquitination and phosphorylation can maintain genomic stability [163]. Histone deubiquitination (DUB) and histone acetylation (HAT) play highly conserved and deterministic roles in eukaryotes. The deubiquitination panel of *Candida glabrata* plays a role in biofilm formation, drug tolerance, and virulence, where biofilm formation and drug tolerance play an important role, and virulence plays a secondary role [164]. Spatial site-blocking induced by histone ubiquitination is capable of disrupting and remodeling the typical structure of nucleosomes, regulating the state of chromatin, assisting in the repair of UV damage, and controlling chromatin segregation. By stimulating the regulation of methylation associated with H3K4 and H3K79, the inhibition of the mutual destruction of nucleosomes occurs [165]. SUMO modifications are small ubiquitin-associated modifiers. SUMO co-adsorbs with a variety of proteins to regulate a wide range of cellular processes: replication, transcription, chromosome segregation, and other aspects. For example, in the nuclear receptor NR4A1, two SUMO ubiquitination modification sites have been identified, and mutation of both sites enhances the stability of this nuclear receptor. SUMOylation and ubiquitination work together to control NR4A1, mediated by RNF4, and interacts with RNFA to control stability [166]. Telomeres act as factors that may cause chromosomal abnormalities and cancer if dysfunction occurs, thus the integrity of telomeres is a very important factor. The integrity and normal function of telomeres require the co-regulation of many factors, and these relevant factors are co-regulated by ubiquitin and SUMO. Crosstalk between these two regulatory pathways has also been reported for DNA repair [167].

Compared with histone methylation and acetylation, other histone modifications are less applied in fungi, and we should continue to study the application of other histone modifications in fungi. With the advancement of technology and the updating of methods, we can gradually explore the functions and regulatory mechanisms of other histone modifications in fungi in depth. This will help us fully understand the gene expression regulatory network of fungi.

## 4. Cross-Regulation of Secondary Metabolism by Epigenetic and Global Regulation

To date, the discovery of novel secondary metabolites has become increasingly difficult, posing a major challenge to researchers. Researchers have therefore shifted their research direction to the intersection of the two: global transcriptional regulation and epigenetic regulation. In secondary metabolic regulatory pathways, some global transcriptional regulators can exist as epigenetic regulators or can influence epigenetic regulation within global transcriptional regulation. This occurs to better regulate secondary metabolite production and is called cross-regulation. Below are some examples.

### 4.1. LaeA

LaeA, as part of the complex with VeA and VelB, is a major positive regulator of fungal secondary metabolic growth [168]. LaeA, as an epigenetic modifier-methyltransferase, is likewise a global transcriptional regulator. With these two modes of regulation, the expression of secondary metabolism-related genes can be regulated by altering the fungal chromosome structure. In *Ganoderma lucidum*, ganoderic acid, as a secondary metabolite, can be regulated by LaeA. Knockdown of the LaeA gene resulted in a significant decrease in mRNA levels and a subsequent dramatic reduction in the output capacity of the secondary metabolite ganoderic acid. Overexpression of the LaeA gene, on the other hand, led to an increase in the concentration of ganoderic acid [169]. LaeA regulates biosynthesis in a wide variety of filamentous fungi, such as *Aspergillus fumigatus* and aflatoxin in *Aspergillus flavus*. Knockout of the LaeA gene also significantly reduced conidial production. In addition, some homologs of LaeA also play an important role in the virulence of secondary metabolites [170]. LaeA was shown to be associated with the regulation of H3K9 methylation levels using global histone methylation analysis, and it was postulated in the study that the putative histone methyltransferase LaeA regulates the methylation levels of histone H3K9 and thus the characterization of the fungus. In the fungus *Penicillium oxalicum*, LaeA was found to be a histone H2B methyltransferase that promotes the methylation of histones H2BK122 and H2BK130 in vitro. In this study, mutations to H2BK122 and H2BK130, on the other hand, resulted in defects in fungal development, reduced conidial production, and broken cellulose hydrolase production. This result is similar to the deletion of LaeA [171]. The production of the secondary metabolite citric acid requires the regulation of the methyltransferase LaeA, and the identification of homologous genes for LaeA in the white-trash fungus *Aspergillus vinelandii* revealed that strains that knocked out LaeA resulted in insufficient production of citric acid. LaeA regulates the methylation levels of histones H3K4 and H3K9 and thus regulates the expression of the CexA gene, which then serves as a regulator of citric acid production [172]. LaeA is both an epigenetic regulator and a global transcriptional regulator, both of which are cross-regulated and, together, play a regulatory role in metabolic pathways.

### 4.2. GcnE

GcnE is an epigenetic regulator involved in acetylation modification, a histone acetyltransferase that transfers acetyl groups to histones and regulates gene expression [173]. The target genes of histone acetyltransferase GcnE include genes for secondary metabolism and asexual development. In the absence of GcnE, the gene for glutamine synthetase is regulated and enzyme activity is increased. A yeast two-hybrid screen of *Aspergillus fumigatus* proteins, along with dual validation by two-fluorescent molecular complementation assays, confirmed the important role of glnA in glutamine synthesis and the silencing of the gene glnA from expression by GcnE in the presence of binding to glnA [174]. According to the literature, GcnE is involved in elevated levels of histone H3K9 acetylation, which affects the synthesis of secondary metabolites. The deletion of the epigenetic regulator GcnE gene resulted in the activation of the production of 12 secondary metabolites in *Aspergillus niger* [175]. GcnE deletion plays a key role in the regulation of fungal asexual development compared to the wild type. GcnE deletion significantly reduces the radial growth of fungal colonies; the number of conidiophores is significantly reduced, and the formation of biofilm mucosa is significantly affected [176]. SAGA/Ada complex-dependent histone acetylation of H3K9 and H3K14 is increased in fungal–bacterial interactions [177]. In *Aspergillus flavus*, GcnE is in the nucleus and regulates cell function. Deletion of GcnE in *Aspergillus flavus* reduces the hydrophobicity of the surface and ultimately the lack of asexual spores, as well as the inability to produce a nucleus, thus inhibiting the growth of *Aspergillus flavus*, and knockout of the GcnE gene would also result in the non-production of aflatoxins by *Aspergillus flavus* [178]. GcnE, an important epigenetic regulator and histone acetyltransferase, acts in conjunction with global regulation in fungal growth and metabolism and is an important novel strategy for altering microbial growth and metabolism and virulence effects [179].

### 4.3. SirE/Hst4, SirB/Hst2

Sirtuin E is a histone deacetylase that deacetylates lysine residues and controls the deacetylation of lysine 9. However, at the same time, Sir E allows global transcriptional regulation from an epigenetic point of view. Sir E is able to exhibit mycelial autolysis, conidial development, synthesis of secondary metabolites, and reduced production of extracellular hydrolases [180]. This NADH+-dependent histone deacetylase regulates histone H3K56 in the nucleus of Aspergillus flavus cells, which is highly sensitive to both DNA damage and redox reactions, illustrating the indispensability of acetylation modifications of histone H3K56 and its biological functions [77]. Four homologs of Sir2 in yeast were identified, namely Hst1, Hst2, Hst3, and Hst4, where Hst1 and Sir2 are a homologous genes. While Hst2 is mainly localized to the cytoplasm, it represses subtelomeric genes and is detrimental to rDNA silencing. Hst3 and 4 act as histone H3K56 deacetylases and are associated with cellular processes and transcriptional silencing. In knockout mutants of Sir2, the promoter is transcribed so that the cohesion proteins are unable to bind and mismatch the rDNA genes [181].

Hst2 can control the rate of mycelial growth in *Ustilaginoidea virens* and will retard mycelial growth rates and reduce viral pathogenicity, while it negatively regulates the biosynthesis of a variety of secondary metabolites and mycotoxins. These mycotoxins include ustilaginoidins, sorbicillin, ochratoxin B, zearalenone, and *O-M*-sterigmatocystin. Thus, we know that it is a global regulatory transcription factor for secondary metabolism. As an epigenetic regulator, it also acts as a histone deacetylase. The two work together to achieve true cross-regulation [182]. Sirtuin HstD/AoHst4 found in *Aspergillus oryzae* is located upstream of LaeA and can regulate fungal secondary metabolism, and it is *A. oryzae* yeast homologue of Hst4. It is involved in fungal development and SM production through LaeA expression. In contrast, the absence of all histone deacetylases in *A. oryzae* suggests that the fungal-specific Sirtuin HstD/AoHst4 is required for the coordination of fungal development and secondary metabolite production [183].

The cross-regulation of epigenetic regulation with global transcriptional regulation has been widely applied to various fungi. It has now become one of the most important research directions in the biosynthesis of natural secondary metabolites in fungi, providing an important research strategy for both fungal mycorrhizal growth, virulence effects, and the extent of secondary metabolite synthesis [5,184].

## 5. Conclusions and Outlook

Fungi use epigenetic mechanisms such as DNA methylation and histone modifications to regulate secondary metabolism and adapt to environmental changes that affect growth, survival, and host adaptation. Specific environments trigger secondary metabolism to produce unique compounds, and epigenetic regulation can prevent fungal infections and influence pathogenicity. However, many unanswered questions remain. In the future, the following areas could be focused on: (1) Refining regulatory networks: deepening the understanding of the specific roles and interrelationships of DNA methylation, histone modifications, and non-coding RNAs in fungal secondary metabolism. (2) Dynamic changes in epigenetic marks: Future studies should focus on how these epigenetic marks change over time and with environmental conditions in order to better understand how fungi adapt their secondary metabolism to different environments. (3) Development of epigenetic tools: In previous studies, by regulating epigenetic modifications in fungi, the production of their secondary metabolites could be altered, which creates the possibility of using epigenetic tools such as CRISPR/Cas9 to optimize the production of secondary metabolites from fungi or to develop new products [185,186]. (4) Specific antifungal strategies: Understanding the relationship between specific epigenetic markers and secondary metabolism may help design antifungal treatments that are more targeted and have fewer side effects. Overall, research on the epigenetic regulation of fungal secondary metabolism has yielded tremendous results, but it is only the tip of the iceberg, and more in-depth and extensive research is still needed. In the future, new technological tools and methodologies could be further developed, and the study of fungal epigenetics could be strengthened by combining multiple approaches, such as genomics, proteomics, and biochemistry, in order to analyze the role of fungal epigenetics at the systemic level, as well as the mechanisms that regulate it internally.

## Figures and Tables

**Figure 1 jof-10-00648-f001:**
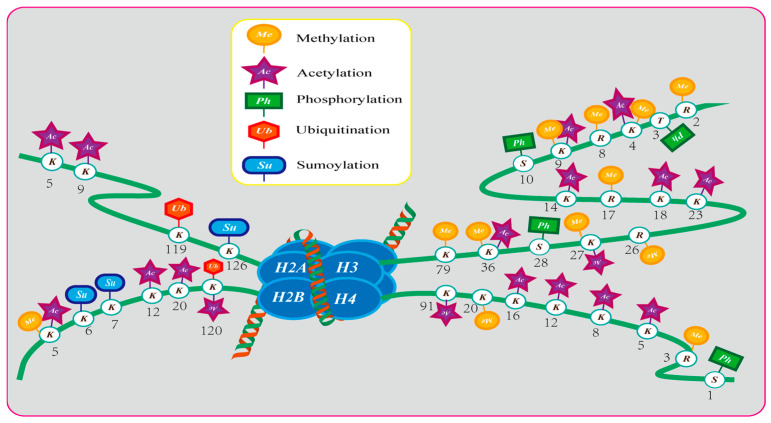
Epigenetic modifications at different sites on histones.

**Figure 2 jof-10-00648-f002:**
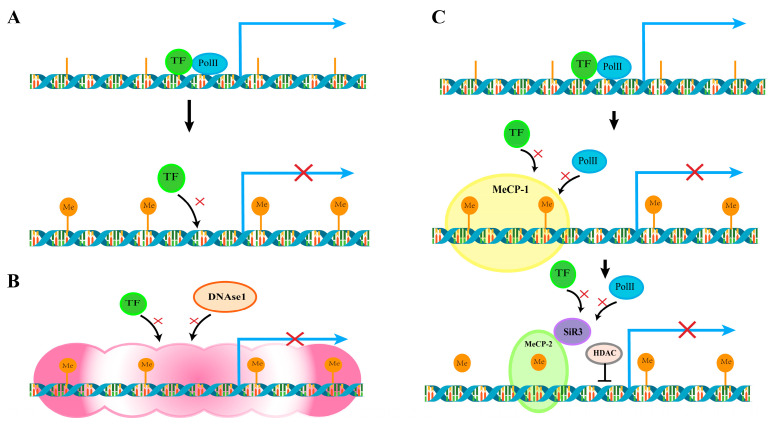
Regulatory mechanisms of DNA methylation: (**A**) Directly preventing transcription factor binding: DNA methylation can prevent transcription factors from binding to DNA, thus preventing the transcription of genes. Transcription factors normally bind to unmethylated DNA sequences to initiate the transcription process of a gene. However, when DNA methylation occurs in the vicinity of a transcription factor binding site, the methylated base pair prevents transcription factors from binding to the DNA, resulting in the repression of gene transcription. (**B**) The formation of inactive chromatin structures can affect gene transcription. (**C**) Methylated DNA can also recruit proteins of the DNA methylation-binding protein family to bind to methylated DNA, forming DNA methylation complexes. These complexes can further interact with co-transcriptional repressor complexes, which together maintain the shutdown state of genes, thereby repressing gene transcription.

**Figure 3 jof-10-00648-f003:**
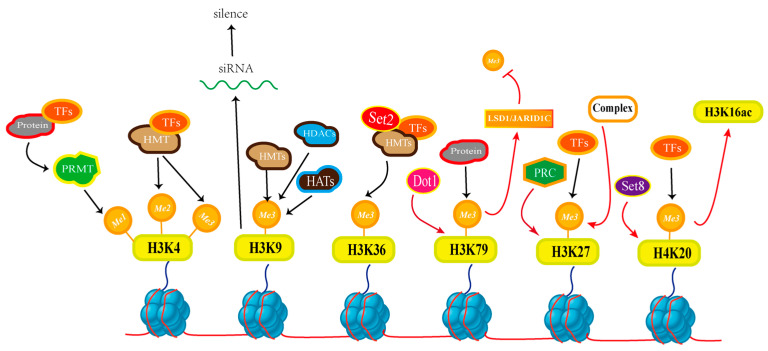
The regulatory process of histone methylation: Gene expression is regulated through the synergistic action of methylation enzymes and demethylation enzymes. Methylases are able to transfer methyl groups to lysine residues of histones, causing methylation of gene regions. The addition of methylated groups leads to the tightening of chromatin, making it difficult for the relevant gene regions to be bound by transcription factors, RNA polymerases and other proteins, thus inhibiting the transcriptional activity of genes.

**Figure 4 jof-10-00648-f004:**
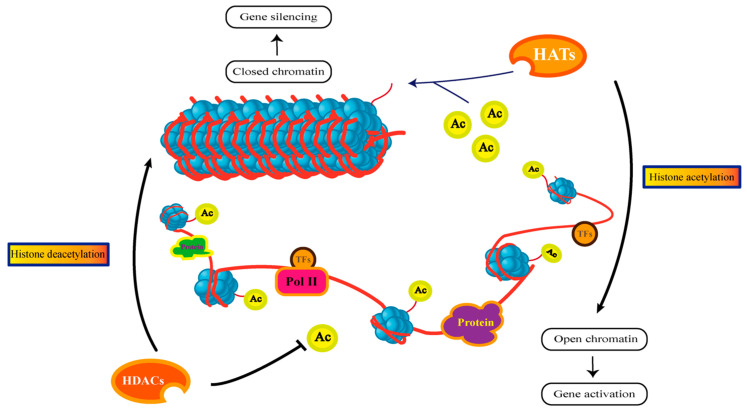
The regulatory process of histone acetylation: Histone acetyltransferases (HATs) transfer acetyl groups to histone proteins for regulation. HATs bind to chromatin-opening-associated proteins and relax chromatin structure by acetylating lysine residues of histones, allowing transcription factors to bind more readily to DNA and facilitating gene transcription. Deacetylases (HDACs) inhibit chromatin acetylation levels by removing acetyl groups from histones. The deacetylation process can further change the structure of chromatin and affect gene transcription. The activity and localisation of HDACs are regulated by many other factors such as cofactors, transcription factors and non-coding RNAs. These regulatory factors can affect the binding and catalytic activity of HDACs and thus the acetylation status of histones.

**Table 1 jof-10-00648-t001:** Overview of epigenetic regulation of secondary metabolism in different fungi.

Species	Epigenetic Type	Secondary Metabolic Effects	Key Genes or Enzymes	Other Physiological	References
*Candida albicans*	DNA methylation	Ergosterol	Lanosterol 14α-demethylase	The repression of gene transcription or expression, and loss of product function	[64,65]
*Cryphonectria parasititica*	DNA methylation	sectored progeny	CpDmt1/CpDmt2	Robust mycelial growth, reduced conidiation, and restricted pigmentation	[66]
*Metarhizium robertsii*	DNA methylation	Regulates energy synthesis and metabolic activity	MrDIM-2/MrRID	Genes with moderately methylated promoter regions are likely to have enhanced transcription	[67]
*Cordyceps* *militaris*	DNA methylation	3′-deoxyadenosine	CmDMTA/CmDIM-2	Methylation modification and DNA recombination can alter a strain’s genotype and thus induce strain degeneration	[68]
*Neurospora crassa*	DNA methylation	Meiosis is silent	DIM-2	Silencing of the transgene as well as its natural homologues	[69]
*Heterobasidion parviporum*	DNA methylation	The expression level of TEs was silenced	SAP-specific genes/NECT-specific gene	saprotrophic growth (SAP) and necrotrophic growth (NECT)	[70]
*Aspergillus flavus*	H3K36me	aflatoxin B1	AshA	Involved in morphogenesis and mycotoxin synthesis	[71]
*Fusarium verticillioides*	H3K36me	FB1 biosynthesis	FvSet2	Defects in vegetative growth, pigmentation, and fungal virulence	[72]
*Colletotrichum higginsianum*	H3K4me	colletochlorins, higginsianins, and sclerosporide	CclA	Significant reductions in virulence and wall penetration ability	[73]
*Aspergillus fumigatus*	H3K4me	gliotoxin	CclA	A slow-growing strain is produced	[74]
*Aspergillus flavus*	H3K14ac/H3K18ac/H3K23ac	aflatoxin B1	MystB	Significant defects in conidiation, sclerotia formation, and aflatoxin production	[75]
*Aspergillus terreus*	H3K27ac/H3K56ac	lovastatin	HstD	Ablation of HstD resulted in decreased mycelial growth, reduced hyphalization, and a significant increase in tylosin biosynthesis	[76]
*Aspergillus flavus*	H4K16ac	aflatoxin B1	MystA	Decreased conidiation, increased sclerotia formation and aflatoxin production	[75]
*Aspergillus niger*	H3K9ac	fumonisin B2	GcnE	Synthesis of more secondary metabolites	[46]
*Aspergillus flavus*	H3K56ac	aflatoxin B1	SirE	Highly sensitive to DNA damage and oxidative stress	[77]
*Alternaria alternata*	H2Bub	Macromolecular complex generation	AaBre1	Mycelial growth, conidial formation and pathogenicity	[78]
*Candida albicans*	H2Bub	antibiotics	Ubp8	Activation of the mycelial program	[79]

## Data Availability

Not applicable.

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
