# Peer review of "Epigenetic Regulation of Fungal Secondary Metabolism"

_jof, 2024, doi:10.3390/jof10090648_

Round 1

Reviewer 1 Report

General comments

The article “Epigenetic regulation of fungal secondary metabolism” reviews the present status of the role of epigenetics in the control of secondary metabolites biosynthesis in fungi. This subject is of interest and novelty. The content of the article is very dense and sometimes repetitive. The major classes of epigenetic factors are described such as DNA methylation/demethylation, histones acetylation/deacetylation, histones methylation/demethylation and other protein modifying factors. However, it is important to make it clear that one thing are the classical pleiotropic transcriptional factors that regulate the biosynthesis of many secondary metabolites at the transcriptional level and a different field is the epigenetic that involves modification of proteins affecting gene expression without altering DNA nucleotide sequences. This is important in the confusion existing about the LaeA factor (see comment below). There are several repetitions in the different sections of the manuscript. That should be avoided. There are also sentences very undetermined with a unclear scientific content (as the sentences indicated in the specific comments below). This should be carefully corrected. Also, some terms need definition to make them accessible for the large scientific audience interested in this field. This reviewer recommends to trim-down the manuscript to improve its quality and to make it more accessible to non-strict specialist in epigenetics and a good improvement in the quality of the english.

Specific comments

1.       Line 26. Comment:  the authors refer to fungi in plural, however in the next sentence they say “It is involved in a” it should be “They are involved in”.

2.       Lines 151-152. The authors say “Transcriptional regulation is one of the very important modes of regulation in the synthesis of secondary metabolites. HbxA is a global regulator of Aspergillus fumigatus that can influence morphogenesis and secondary metabolism”.

Comment:  Avoid to start phrases with unknown numbers (HBxA); it would be better to indicate “The global regulator of A fumigatus HBxA…” and then indicate what type of regulator it is

3.       Lines 171-172 “Epigenetic modifications, which occur after the translation of proteins, have an effect on the packaging of chromatin, which subsequently affects the transcriptional machinery of DNA, which in turn affects gene expression [45]”.

Comment: This sentence is too unspecific. Not all proteins modify posttranlationally are involved in epigenetic. Many acetylation/deacetylations are involved in general protein modifications unrelated to epigenetics.  The authors should clarify that epigenetics occur after posttranslational modification of specific sets of proteins by DNA methylation, acetylations and deacetylation that are involved in the control of DNA expression

4.       Lines 208  “Application of chemical epigenetic modifiers to the plant endophyte fungus Aspergillus fumigatus was able to significantly alter metabolic profiles, generate a variety of new natural products, and isolate immunosuppressive agents [54].”

Comment: The authors refer to similar aspects of A. fumigatus in several places in the article. Please concentrate all the information of the most representative filamentous fungi (e.g. A. fumigatus) in the adequate review section to avoid repetition

5.       Lines 246-247. “DNA methylation of C. chrysogenum can induce strain degeneration and have important effects on”.

Comment: Do you means P. chrysogenum or C. acremonium?. Anyway reference 64 do not correspond to this fungi. Please, clarify.

6.       Lines 419 (Legend of fig 3). It says:” The regulatory process of histone methylation: Gene expression is regulated through the synergistic action of methylation enzymes and methylation de-enzymes”.

Comment:  What are de-enzymes? I guest the authors refer to demethylation enzymes?.

7.       Lines 438-444 The authors state “That is, transcriptional regulation of gene expression is dependent on the state of chromatin, and chromatin competence itself is dependent on epigenetic regulation [132]. And acetylation modifications are also present as a key factor in regulating memory processes. transcription of DNA is necessary for the development and maintenance of long-term memory capacity and can cause a range of behavioral changes. For example, the inhibition of HDACs enhances long-term memory and HAT inhibition impairs memory [133].

Comment: Do the authors refer to filamentous fungi memory ?, if they refer to the memory of mammals indicate it clearly and specify which type of mammals. The word behavioral is misspelled.

8.       Lines 463  “There are many filamentous fungi that are capable of synthesizing excellent secondary metabolites beneficial to human growth during microbial growth”

Comment: This sentence is repetitive to one at the beginning of the review. Please, avoid repetition the same sentence. In addition the sentence  is nonsense as it is written. Perhaps you refer that “filamentous fungi during their growth synthesize excellent secondary metabolites beneficial to human health”. The word remodeling is misspelled .

9.       Line 533  that authors state “Histone acetylation modifications have been extensively explored in fungi, but in order to more comprehensively understand the functions and regulatory mechanisms of histone acetylation in fungi, it is necessary to further develop new technological tools and methods, as well as to strengthen the study of dynamic regulation, and to combine multiple approaches, such as genomics, proteomics, and biochemistry, in order to analyze the roles and regulatory mechanisms of histone acetylation at the systemic level

Comment: This paragraph fits better in the section of future perspectives at the end of the manuscript

10.   Lines 624-632  “The production of secondary metabolites is controlled by a complex regulatory network at multiple levels in both eukaryotes and prokaryotes, ranging for example from pathway-specific regulation to epigenetic regulation to global regulation [164]. There is an abundance of biosynthetic gene clusters (BGCs) on the fungal genome, while 90% of these clusters are unexplored. Knockdown or overexpression of genes in biosynthetic gene clusters can have a significant impact on the pro-duction of secondary metabolites. This is especially true for the manipulation of global regulators or for the manipulation of epigenetic factors [165]. Global transcriptional regulators are factors that are able to regulate the expression of genes throughout the genome or over a large area.

Comment: This is redundant with the same presentation made in section I where the authors discuss the organization of the secondary metabolite gene clusters. Please avoid repetitions.

11.   Lines 634 635 It states “Epigenetic regulators are modifications that can affect the chromosomal structure of a gene or modulate gene expression without changes at the DNA level, such as chromatin remodeling complexes

Comment: The authors define again the concept of epigenetics; this is repetitive of the same definition present in the first part of the manuscript. Please delete. The word remodeling is uncorrectily written.

12.   Lines 654-670 “The methyltransferase LaeA, associated with the regulation of methylation aspects of histone H3K9, regulates fungal metabolic functions based on the fact that it is an epigenetic factor. It regulates mycelial growth, glucose consumption, and the involvement of LaeA in cellulose expression. Some negative growth factors in fungi are also able to be regulated by LaeA, such as the global regulator McrA Relevant molecular manipulation of the LaeA gene is an emerging production tool in the use of metabolic engineering to improve strain fermentation [166,169]. The secondary metabolite peptides of the biocontrol fungus, Trichoderma longibrachiatum SMF2, have a wide range of biological activities, and deletion of T/LAE1, a homologue of LaeA in Trichoderma longibrachiatum SMF2, has a down-regulation of secondary metabolite production and mycelial growth effects. It has also been hypothesized to be a global regulatory transcription factor [170].

Comment: This paragraph refers to the regulator LaeA in fungi. Scientifically the mechanism of action of LaeA is still unclear particularly that LaeA acts as an epigenetic factor. It is demostrated that LaeA is a pleiotropic regulator that controls the biosynthesis of many secondary metabolites in fungi and there are some key references that should be cited but although LaeA has an S-adenosylmethionine recognizing sequence there is no absolute evidence of the methylation role of LaeA on epigenetic control of secondary metabolites. This is well documented in articles as Özlem Sarikaya-Bayram et al (2015) and Patanaman et al (2013 in JBC).. Importantly the references cited by the authors 166 to 171, all describe the putative methyltransferase role of LaeA on the biosynthesis of secondary metabolites.  please note that the putative name of methyltransferase is not an epigenetic methyltransferase as described in the cited articles. The authors should refocus the section of LaeA.

13.      Lines 712-713 It states “Hst2 can control the rate of mycelial growth in Ustilaginoidea virens and will retard mycelial growth rates and reduce viral pathogenicity, while it negatively regulates the biosynthesis of a variety of secondary metabolites, mycins, and toxins.

Comments: Please, explain clearly what the authors means by the term mycins

14.   Lines 745-746 It states:” 3) Development of epigenetic tools: according to our study,…

Comment: If the authors refer to their own study indicate reference, otherwise delete the word refering self citation

Reviewer 2 Report

Yufei Zhang and colleagues present a review article on fungal secondary metabolism, emphasizing on the regulation at the epigenetic level. The article is comprehensibly written and contains interesting information for people working in this field, but also beyond. The references cited are appropriate and the figures are informative and nicely prepared.

In my opinion, the article would really benefit from some shortening and compacting, without compromising the readability. Some examples are given below, but there are also other cases throughout the text where sentences could be merged and shortened.

Lines 18-20: This information is also given above and these sentences could be omitted.

Lines 27-31: These two sentences contain similar information.

The same also for lines 27-31, 42-47, etc, etc.

Lines 48: This information on toxins contained in fungi that are at the same time used as biological controls is very important and many times underestimated. The authors may want to consider highlighting this issue further.

Line 58: Fungal consortia have also proved important between fungi and bacteria. Authors may want to consider including some additional information here.

Line 139: Interspecies communication has also been shown in Aspergillus and also in mycorrhizal fungi. Perhaps the authors would like to include this information somewhere appropriate.

Chapter 5 could also be more compacted.

Line 68 and 290: Please use plural for ‘role’ and ‘mechanism’.

Round 2

Reviewer 1 Report

The authors have reply satisfactorily to most of the reviewer questions, except question 4. That requested modification would greatly improve the manuscript avoiding repetitions

No detailed additional comments